# Architect: A tool for aiding the reconstruction of high-quality metabolic models through improved enzyme annotation

**Nirvana Nursimulu**[1,2], **Alan M. Moses**[1,3], **John Parkinson**[2,4,5]*

**1** Department of Computer Science, University of Toronto, Toronto, Ontario, Canada, **2** Program in Molecular Medicine, The Hospital for Sick Children, Toronto, Ontario, Canada, **3** Department of Cell & Systems Biology, University of Toronto, Toronto, Ontario, Canada, **4** Department of Molecular Genetics, University of Toronto, Toronto, Ontario, Canada, **5** Department of Biochemistry, University of Toronto, Toronto, Ontario, Canada

* john.parkinson@utoronto.ca

**Data Availability Statement:** Architect's code can be found at https://github.com/parkinsonlab/Architect, with additional information at https://compsysbio.org/projects/Architect. Enzyme

## Abstract

Constraint-based modeling is a powerful framework for studying cellular metabolism, with applications ranging from predicting growth rates and optimizing production of high value metabolites to identifying enzymes in pathogens that may be targeted for therapeutic interventions. Results from modeling experiments can be affected at least in part by the quality of the metabolic models used. Reconstructing a metabolic network manually can produce a high-quality metabolic model but is a time-consuming task. At the same time, current methods for automating the process typically transfer metabolic function based on sequence similarity, a process known to produce many false positives. We created Architect, a pipeline for automatic metabolic model reconstruction from protein sequences. First, it performs enzyme annotation through an ensemble approach, whereby a likelihood score is computed for an EC prediction based on predictions from existing tools; for this step, our method shows both increased precision and recall compared to individual tools. Next, Architect uses these annotations to construct a high-quality metabolic network which is then gap-filled based on likelihood scores from the ensemble approach. The resulting metabolic model is output in SBML format, suitable for constraints-based analyses. Through comparisons of enzyme annotations and curated metabolic models, we demonstrate improved performance of Architect over other state-of-the-art tools, notably with higher precision and recall on the eukaryote *C. elegans* and when compared to UniProt annotations in two bacterial species. Code for Architect is available at https://github.com/ParkinsonLab/Architect. For ease-of-use, Architect can be readily set up and utilized using its Docker image, maintained on Docker Hub.

## Author summary

An organism's growth and survival are largely guided by its ability to synthesize crucial metabolites like amino acids and ribonucleotides from such compounds as water, glucose

annotation results and reaction database information can be found at https://compsysbio.org/projects/Architect/Database. Organism-specific sequences are found under their respective directories at https://compsysbio.org/projects/Architect/Architect_reconstructions. Architect's Docker image is maintained at https://hub.docker.com/r/parkinsonlab/architect.

**Funding:** This work was supported by grants from the Canadian Institute for Health Research grant (PJT-152921) and the Natural Sciences and Engineering Research Council (RGPIN-2019-06852) to JP (https://cihr-irsc.gc.ca; https://www.nserc-crsng.gc.ca). NN was supported by a SickKids RestraComp scholarship (https://www.sickkids.ca/en/research/research-training-centre/scholarships-fellowships-awards). Computing resources were provided by the SciNet HPC Consortium (https://www.scinethpc.ca); SciNet is funded by: the Canada Foundation for Innovation (https://www.innovation.ca) under the auspices of Compute Canada (https://www.computecanada.ca); the Government of Ontario (https://www.ontario.ca); Ontario Research Fund–Research Excellence (https://www.ontario.ca/page/ontario-research-fund) and the University of Toronto (https://www.utoronto.ca). The funders had no role in study design, data collection and analysis, decision to publish, or preparation of the manuscript.

**Competing interests:** The authors have declared that no competing interests exist.

and nitrogenous molecules like ammonia. Accurate knowledge of such biochemical reactions—catalyzed by enzymes encoded within the genome—can advance our understanding of disease drivers as well as guide attempts at engineering strains of bacteria with desired metabolic capacities. While biochemical experiments can accurately characterize metabolism, these are time-consuming at genome-scale. Instead, genome-scale metabolic models can be computationally built then iteratively refined through comparisons of *in silico* simulation results and biochemical experiments. Here, we describe Architect, a method for automatic enzyme annotation and metabolic model reconstruction. Our tool leverages the strengths of existing enzyme annotation tools to predict the biochemical capacities of an organism and then uses these predictions to build a simulation-ready metabolic model. We find that Architect produces more accurate enzyme annotations than the individual tools, as well as higher-quality metabolic models compared to other automatic metabolic model reconstruction tools. We provide Architect to the metabolic modelling community in the hope that it may facilitate the transition from knowing an organism's encoded sequences to an understanding of its metabolic capacities.

This is a *PLOS Computational Biology* Methods paper.

## 1. Introduction

Metabolic modeling has been used for engineering strains of bacteria for bioremediation, for understanding what drives parasite growth, as well as for shedding light on how disruptions in the microbiome can lead to progression of various diseases [1–3]. In any of these applications, the standard protocol is to first construct an initial draft of the metabolic model of the organism(s) (consisting of the biochemical reactions predicted present) followed by a gap-filling procedure, whereby additional reactions are introduced to ensure that simulations can be performed [4]. Importantly, errors introduced at any steps of model reconstruction can impact downstream simulations and result interpretation [5]. For instance, false positive enzyme predictions may mask the essentiality of key pathways; on the other hand, the organism's metabolic abilities may be underestimated when metabolic enzymes and pathways are incorrectly left out or under-predicted [6]. While these concerns can be addressed through dedicated manual curation, such efforts tend to be extremely time-consuming. Instead, attention has turned to the use of automated methods, such as PRIAM, CarveMe and ModelSEED, the former able to automatically annotate enzymes from sequence and the latter two capable of generating fully functional genome-scale metabolic models [7–9]. Given a genome of interest, CarveMe uses sequence similarity searches to assign confidence scores to reactions within a universal model of metabolism. Based on these scores, a genome-specific metabolic model is then reconstructed by removing reactions that are either not identified or poorly supported, and adding in reactions to fill gaps to construct functional pathways [7]. On the other hand, ModelSEED relies on an initial annotation of sequences by RAST (Rapid Annotation using Subsystem Technology) to produce a draft model, following which gap-filling is performed to enable biomass production in either complete or user-defined media [9].

A key step in this process is the accurate identification of enzymes based on sequence data alone and can be formally defined as follows: given an amino acid sequence, what are its associated enzymatic function(s), if any? The problem is a multi-label classification problem; here

we consider enzymatic functions as defined by the Enzyme Commission (EC), in which enzymes are assigned to EC numbers representing a top-down hierarchy of function [10]. Enzyme annotation can be performed by inferring homology to known enzymes based on sequence similarity searches using methods such as BLAST and DIAMOND [11,12]. However, such methods do not consider the overlap of sequence similarity between enzyme classes and are prone to an unacceptable rate of false positive predictions [13]. To overcome such errors, a number of more specialized tools have been developed that take advantage of sequence features or profiles specific to individual enzyme classes [8,13–15]. For example, DETECT (Density Estimation Tool for Enzyme ClassificaTion) considers the effect of sequence diversity when predicting different enzyme classes [13,14], while PRIAM and EnzDP rely on searches of sequence profiles constructed from families of enzymes [8,15]. Each tool provides different advantages in terms of accuracy and coverage.

Here we present Architect, a tool capable of automatically constructing a functional metabolic model for an organism of interest, based on its proteome. At its core, Architect exploits an ensemble approach that combines the unique strengths of multiple enzyme annotation tools to ensure high confidence enzyme predictions. Subsequently gap-filling is performed to construct a functional metabolic model in Systems Biology Markup Language (SBML) Level 3 format [16] which can be readily analysed by existing constraints-based modeling software. Commensurate with its name, Architect not only designs the metabolic model of an organism, but it also coordinates the sequence of steps that go towards the SBML output given user specifications, such as the definition of an objective function for gap-filling. We evaluate the performance of Architect both in terms of its ability to perform accurate enzyme annotations, relying on UniProt/SwissProt sequences as a gold standard [17] and, separately, as a metabolic model reconstruction tool by focusing on 3 organisms for which curated metabolic models have already been generated (*Caenorhabditis elegans* [18], *Neisseria meningitidis* [19] and *E. coli* [20]). Compared to other state-of-the-art methods [7–9], Architect delivers improved performance in terms of enzyme annotation and can predict phenotype with comparable accuracy with the top performers given a high quality reaction database. While Architect does not predict cellular localization, an important aspect of eukaryotic metabolic models, its generation of a metabolic model with improved annotations may facilitate the modeler's transition from protein sequences to a functional metabolic model. Finally, to ease set-up and deployment, we maintain a Docker image encapsulating Architect's numerous dependencies on Docker Hub; Architect's dedicated GitHub repository contains detailed instructions for running Architect using Docker [21,22].

## 2. Results

### 2.1 Ensemble methods improve enzyme annotation

The motivation for developing an ensemble enzyme classifier is driven by the hypothesis that different enzymes (as defined by EC numbers) may be better predicted by different tools and, hence more accurate annotations may be obtained by combining predictions from individual tools. Based on this hypothesis we developed a novel enzyme prediction and metabolic reconstruction pipeline we term Architect (**Fig 1**). In brief, the pipeline begins with the prediction of enzyme annotations from proteome data (Module 1) using an ensemble classifier that combines predictions from five enzyme annotation tools (DETECT, EnzDP, Catfam, PRIAM and EFICAz; [8,13–15,23,24]). Next, these predictions of enzyme classification numbers are used to construct a functional metabolic model capable of generating biomass required for growth (Module 2).

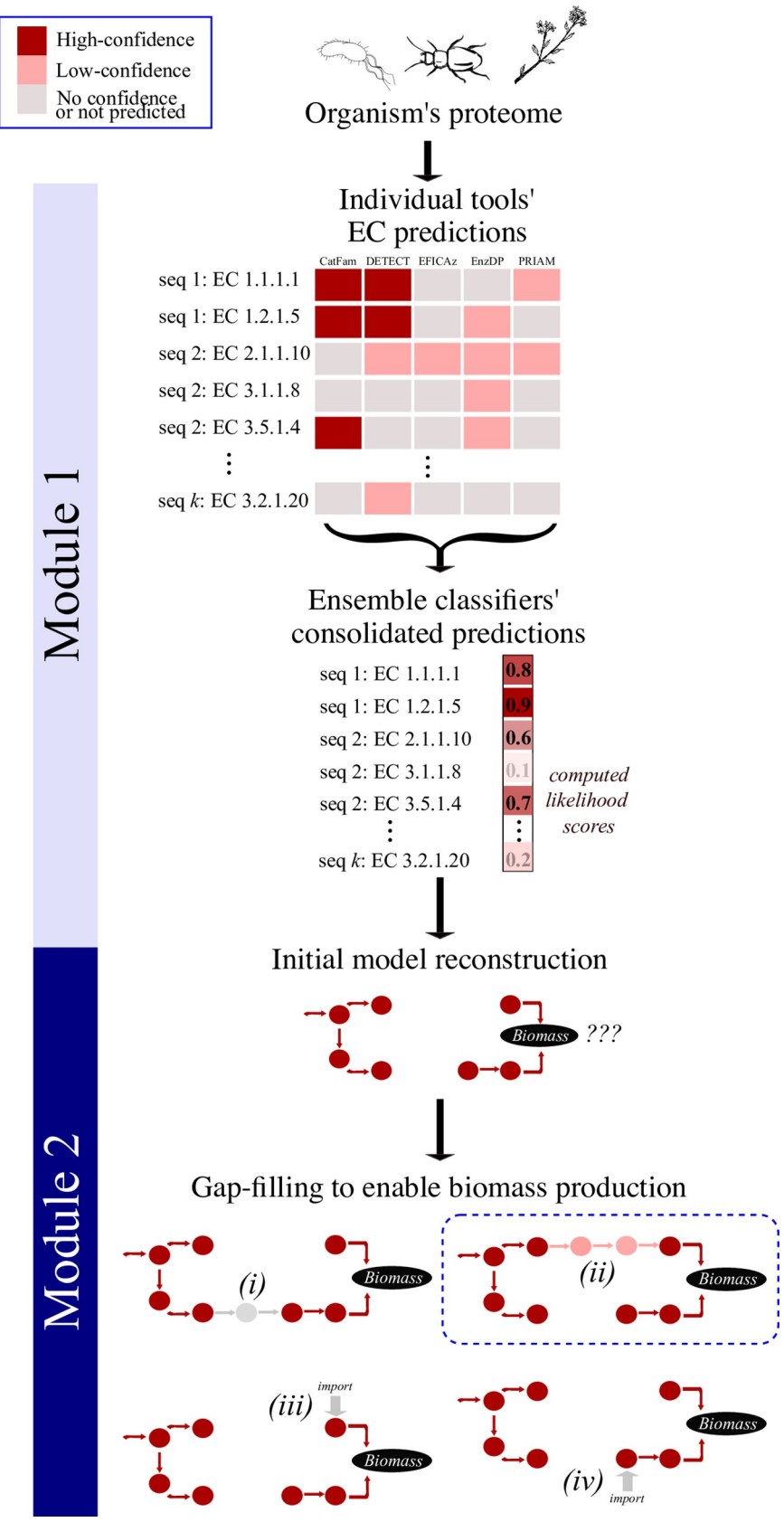

**Fig 1. Overview of Architect's methodology.** Given an organism's protein sequences, Architect first runs 5 enzyme annotation tools, then computes a likelihood score for each annotation using an ensemble approach (module 1). From high-confidence EC predictions, Architect reconstructs a high-confidence metabolic model which it then gap-fills to enable biomass production using the aforementioned confidence scores (module 2). In the illustrated example, 4 sets of reactions are considered for gap-filling, with the solution highlighted in the blue box yielding the highest score.

To test our hypothesis, we compared the performance of individual tools of interest and of various ensemble methods on a dataset of enzymatic sequences. Here we investigated three methods (naïve Bayes, logistic regression and random forest) and employed five-fold cross-validation in which ensemble methods were trained on 80% of the enzymes annotated in SwissProt. Once trained, the performance of each classifier was tested on the remaining annotated enzymes by evaluating their high-confidence predictions against the database's annotations. In addition to the classifiers and individual tools, we also examined the performance of a 'majority rule' approach (in which we assign an EC label to a protein on the basis of voting among the five tools), as well as an 'EC-specific best tool' approach (in which we assign an EC label to a protein based on best performing tools for that EC as seen in training). The performance of each dataset was computed using macro-averaged precision, recall and F1 to ensure that smaller EC classes were equally represented; performance on the non-enzymatic dataset (i.e. protein sequences not associated with either complete or partial EC annotations) is computed using specificity (see **S1 Text**).

Overall, we found that, with the exception of the majority rule, ensemble methods outperformed individual tools, resulting in both higher precision and recall (**Fig 2 and S1 Table**). For example, the highest precision and recall of the individual tools—obtained by DETECT and PRIAM respectively—are lower than most of the ensemble methods applied. Indeed, except for majority rule, most ensemble methods perform similarly on the entire test set, as well as on subsets of test sequences with lower sequence similarity to training sequences (**S1 Fig**). Additionally, macro-recall on multifunctional proteins is decreased for the naïve Bayes, logistic regression and random forest classifiers when applying a heuristic which filters out predicted ECs other than the top-scoring EC and frequently co-occurring enzymes as seen in the training data (**S2 and S3 Figs and S1 Text**); therefore, henceforth, we evaluate performance of these classifiers by considering all their high-confidence EC predictions.

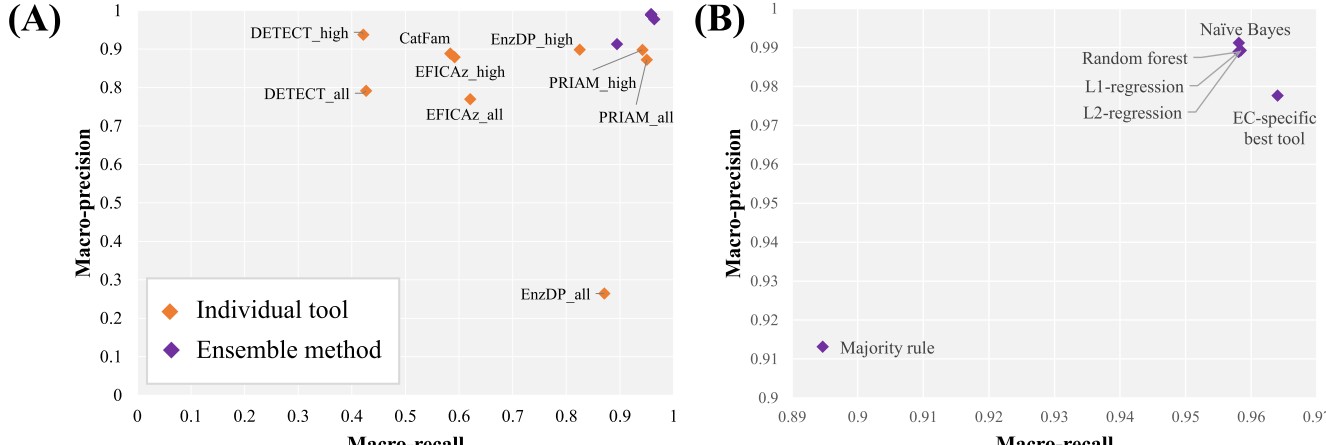

**Fig 2. Performance of individual and ensemble enzyme annotation tools.** (A) Precision/recall graph indicating performance of each method on the enzymatic test set from SwissProt, (B) focus on the improved performance of the ensemble methods. Our results show that combining predictions using almost any ensemble method gives better performance than using any individual tool.

Next, we consider the possibility that higher predictive range (defined as the number of ECs that a tool can predict) primarily drives the increased performance of the ensemble methods. Indeed, the ensemble approaches are superior when quantifying performance on ECs predictable by at least 2 tools (**S4 Fig**) but have similar precision and recall as DETECT on sequences annotated with ECs predictable by all tools (**S2 Fig**). However, when looking at DETECT's or PRIAM's class-by-class performance on ECs that they can predict, the ensemble method has higher precision and recall on more ECs than either DETECT (better precision on 176 versus 18 ECs and better recall on 116 versus 37 ECs) or PRIAM (better precision on 98 versus 38 ECs and better recall on 191 versus 41 ECs) (**S5 Fig**).

Given the main application of Architect is to annotate enzymes to an organism's proteome, we were interested in assessing the ability of the ensemble approaches to minimize false positives. Applied to a set of proteins without EC annotations in SwissProt, we found that only the Naïve Bayes classifier gave comparable specificity as the individual tools (**S6 Fig**). Given the slightly elevated performance in terms of precision (for the enzymatic dataset) and specificity (for the non-enzymatic dataset), we chose the naïve Bayes classifier as the preferred method for Architect. We next investigated the performance of Architect to annotate the proteomes of three well-characterized organisms (*C. elegans*, *N. meningitidis* and *E. coli*; **S7 Fig** **and S2 Table**). We consider those annotations that feed into Architect's model reconstruction module and compare them against high-confidence predictions by DETECT, EnzDP and PRIAM alone, these tools chosen due to their performance on the enzymatic dataset. For all three species, Architect yields both higher precision and recall than DETECT, and higher recall than EnzDP. In *C. elegans*, Architect gives higher recall than either PRIAM or EnzDP, albeit at the expense of precision and specificity. Overall, these results demonstrate Architect's wider applicability to annotate specific organisms.

Finally, we investigated whether combining predictions from all 5 tools is required to obtain improved performance with respect to the individual tools. To this effect, we built the naïve Bayes-based method using predictions from fewer tools, then calculated performance once again on the held-out test set (**S8 Fig**). We observe that this procedure has a greater impact on macro-recall than macro-precision. In particular, leaving out predictions from both tools with the highest predictive ranges (EnzDP and PRIAM) had the greatest impact, while the F1-score decreased least when the tools with the lowest predictive ranges (CatFam and DETECT) are not included in the classifier. We also find that different tools are complementary to each other. While performance is mostly unaffected by excluding predictions from any single tool, combining predictions from at least 2 tools improves performance compared to using any single tool's predictions. Indeed, simply combining predictions from PRIAM and any other tool yields better macro-precision than any tool in isolation. Intriguingly, training on predictions from EnzDP and PRIAM results in the highest performance among pairs of tools, with macro-precision showing only minimal increases with the inclusion of any other tool's predictions. These results suggest that a user may obtain reasonably improved performance by combining predictions from fewer than 5 tools, for example, by excluding tools with longer running times (e.g. EFICAz [25]).

## 2.2 Automated metabolic reconstruction using Architect

In addition to predicting suites of enzymes from an organism's proteome, Architect utilizes these predictions to automatically reconstruct a functional metabolic model capable of generating biomass required for growth (see **Methods**). The process begins by querying the set of EC activities predicted by module 1, against a database of known reactions (either the Kyoto Encyclopedia of Genes and Genomes; KEGG [26] or the Biochemical, Genetic and Genomic

knowledgebase; BiGG [27], to construct an initial high confidence metabolic network). In each case, the reaction databases have undergone some level of curation to ensure the appropriateness of the chemical reactions being modelled ([7] and section D of **S1 Text**). Next a gap filling algorithm is applied to identify enzymes absent from the model (potentially arising from uncharacterized enzymes or sequence diversity [28,29]) to ensure pathway functionality and the ability of the model to generate biomass. Given that biomass functions are organism- and possibly condition-specific [30], users are required to specify a biomass function for the purposes of gap-filling.

To evaluate Architect as a reconstruction tool, metabolic models for the three species previously investigated (*C. elegans*, *N. meningitidis* and *E. coli*) were generated by Architect and three similar tools, CarveMe [7], PRIAM [8] and ModelSEED [9]. CarveMe performs metabolic model reconstruction in a top-down manner, retaining in a final functional model those reactions from the BiGG database [27] predicted with higher confidence scores for the organism of interest and required for model functionality; the confidence score is by default computed based on sequence similarity [12]. On the other hand, PRIAM uses its high-confidence predictions to output a metabolic model based on KEGG; since the database is automatically downloaded and thus devoid of any curation, PRIAM models are not simulation-ready. Last, ModelSEED produces a draft model from annotations made using RAST, which it then gap-fills to enable biomass production given specific substrate availabilities or under the assumption of complete media. To account for differences that may be introduced from using different databases of reactions, two versions of Architect models were predicted using either the KEGG or the BiGG database. Furthermore, the same biomass reactions as used by CarveMe were used for BiGG-based Architect reconstructions and individual BiGG identifiers translated to KEGG identifiers for KEGG-based reconstructions.

In general, we observe that more genes are represented in models generated using Architect with BiGG reaction definitions compared to those generated using KEGG reactions, or when using CarveMe or ModelSEED (**S3 Table**). This difference in gene representation in Architect models generated using the different reaction databases may be a consequence of the use of sequence similarity to identify non-EC related reactions. At the same time, both Architect models associate to reactions the most genes from *C. elegans*, highlighting the utility of the tool for eukaryotic reconstructions. Next, we observe the high number of gap-filling reactions added by CarveMe compared to Architect to enable growth of *N. meningitidis* and *E. coli* in minimal media with [31,32] and without the presence of oxygen respectively. In particular, this procedure introduced 2,774 and 1,510 reactions in *N. meningitidis* and *E. coli*, intriguingly leading to the prediction of the same set of reactions in both (thus only differing in terms of gene-protein-reaction associations). The prediction of the same set of reactions by CarveMe may be due to its reaction database's focus on primary metabolism [7], thereby possibly biasing the algorithm towards selecting the same set of reactions. At the same time, CarveMe's models have the highest percentage of unblocked reactions. This is likely a consequence of CarveMe's top-down approach, emphasizing, prior to gap-filling, the creation of a gapless model unlike the bottom-up nature of Architect and ModelSEED.

To validate Architect's model reconstruction strategy, we next compared the output of the reconstruction tools to previously curated metabolic models [18–20]. Focusing on the KEGG-based reconstructions (**Figs 3 and S9**), Architect produced models of higher precision than either PRIAM or CarveMe for *C. elegans* and *N. meningitidis*, and higher recall for *C. elegans* alone. However, in *E. coli*, CarveMe and ModelSEED have significantly better precision and CarveMe better recall than Architect, likely a result of the presence of curated *E. coli*-related reactions in the BiGG and ModelSEED databases. Similarly, Nmb_iTM560 was based on the *i*AF1260 *E. coli* model [33], again likely contributing to each tool's higher recall when

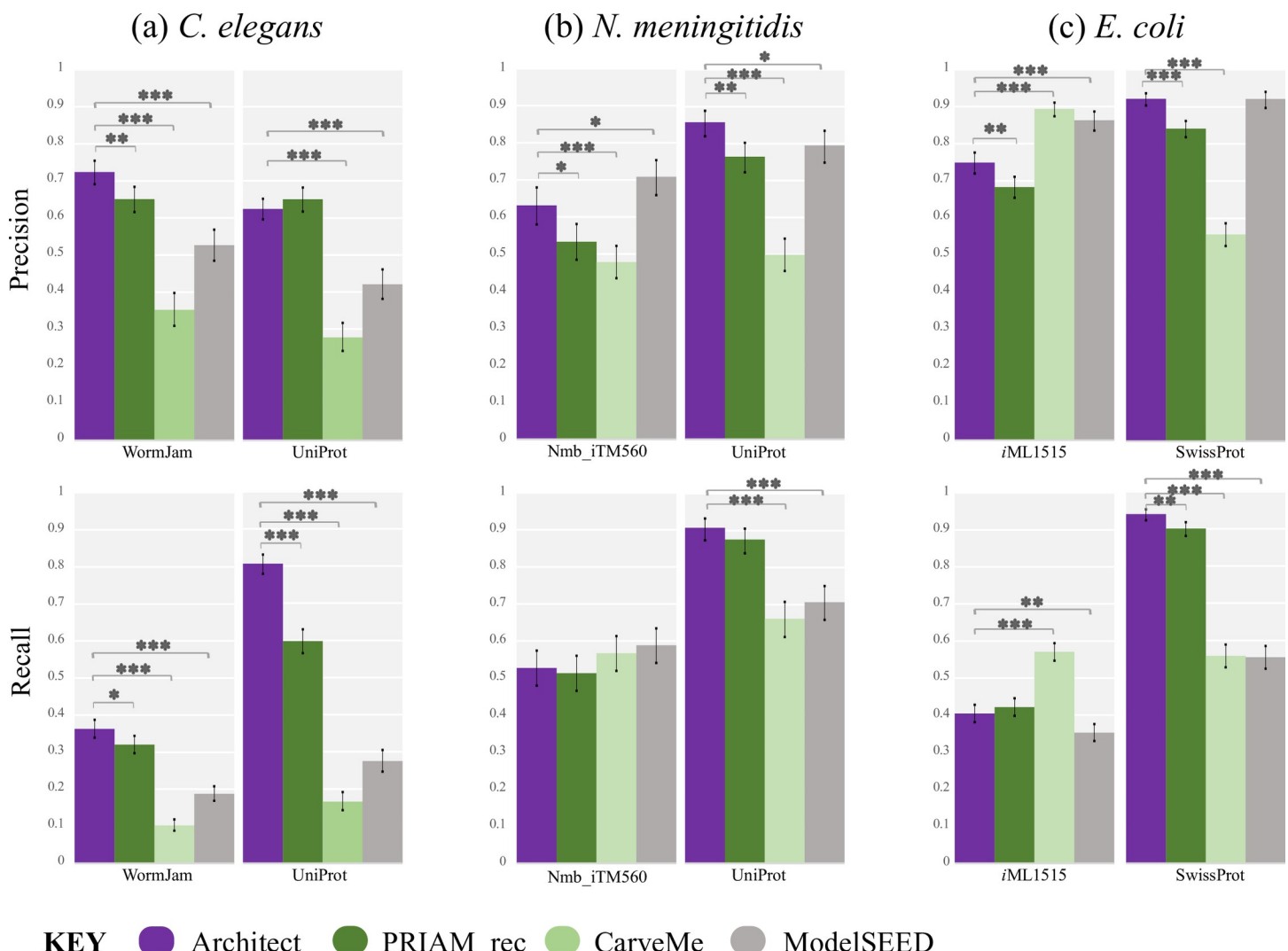

**Fig 3. Performance of Architect as a model reconstruction tool versus CarveMe and PRIAM (as a model reconstruction tool).** Quality of annotations is computed against the curated models over the genes found in these models, and against UniProt/SwissProt when restricting to those sequences found in the database and with ECs present in the KEGG reaction database. SwissProt was used for *E. coli* as the sequences were all present in the curated database. Error bars show the 95% confidence interval for precision and recall, each considered as the estimate of a binomial parameter. P-values, computed using Fisher's exact test, are calculated only between Architect and either CarveMe or PRIAM (with *, ** and *** representing *p* less than 0.05, 0.005 and 0.0005 respectively).

reconstructing a *N. meningitidis* metabolic model. At the same time, ModelSEED's higher precision in *N. meningitidis* can also be attributed in part to a lower EC coverage of true positives in the Architect model, with 33 of the 49 true positive annotations by ModelSEED alone concerning ECs not covered by Architect's KEGG database; this suggests that ModelSEED's reaction database may be useful for Architect reconstructions. Interestingly, when the models are compared against organism-specific datasets retrieved from UniProt, Architect has higher precision and recall than either CarveMe or PRIAM for *E. coli* (as well as higher precision and recall than CarveMe and ModelSEED for the other two species). This highlights differences in annotations associated with the curated models and UniProt. We also note that higher EC coverage in KEGG compared to BiGG (**S10 Fig**) may contribute towards higher recall by Architect and PRIAM compared to CarveMe, a factor we account for by next using the BiGG database for Architect's model reconstructions.

Turning to models constructed with the BiGG database, as for the KEGG-based models, we find that Architect has higher precision for both *C. elegans* and *N. meningitidis*, and higher recall for the former (**S11 and S12** Figs). However, we now observe similar precision in *E. coli*, consistent with the reliance on the BiGG database, which avoids the inclusion of ECs exclusive to the KEGG database (and hence absent in *i*ML1515). As expected from the lower coverage of ECs provided by the BiGG database, recall differences in *E. coli* qualitatively remain unchanged while in *N. meningitidis*, Architect's recall, sacrificed for higher precision, is now significantly lower. Again, to account for the construction of the BiGG database from previously curated metabolic reconstructions that include *E. coli*, we compared the protein-EC annotations in the reconstructed models to those found in UniProt/SwissProt. Architect is then perceived as having higher precision in both *N. meningitidis* and *E. coli*, and greater recall in *E. coli*. Thus, our results indicate that Architect may be used to produce models with more accurate EC annotations than either CarveMe, the PRIAM-reconstruction tool or Model-SEED. Additionally, the choice of reaction database when running Architect may impact the range of ECs covered. Indeed, it should be noted that the BiGG reaction database used here contains reactions from bacterial models only; thus, use of KEGG for eukaryotic reconstructions is more appropriate, as evidenced by the better recall with the *C. elegans* metabolic model.

Notwithstanding these findings, annotations made by Architect and PRIAM are based on cutoffs; thus, high-confidence enzyme—and therefore—reaction predictions are dependent on the threshold specified. To study the impact of such parameter specification, 20 Architect and PRIAM models were reconstructed at several varying cutoffs (**S13, S14 and S15** Figs). As expected, more stringent cutoffs lead to predictions of gene-EC annotations with lower recall but higher precision. However, these changes overall have little impact on the main findings of this study, suggesting, in the case of Architect for instance, that most of its predictions are already made with high-confidence.

In addition to these benchmarking studies, we assessed the quality of each reconstruction using MEMOTE [34], a tool which runs a series of tests to score a metabolic model by such metrics as the level of annotation of reactions and metabolites and stoichiometric consistency. We find that Architect models reconstructed using KEGG reaction definitions achieve higher overall MEMOTE scores than those reconstructed using either CarveMe, PRIAM or Model-SEED largely due to the presence of additional information describing reactions and metabolites (**S4 Table** and **S1 Data**). This indicates that models output by Architect may be more interpretable to users, a helpful advantage especially when additional organism-specific data need to be integrated to the output model.

## 2.3 Metabolic reconstructions benefit from annotation tools with high predictive range

From the previous comparisons of metabolic reconstructions, it is clear from the difference in precision and recall that there is a difference between Architect's performance as an enzyme annotation tool and as a tool for model reconstruction. This raises the question of whether improvement in enzyme annotation is associated with a corresponding improvement in accuracy of model reconstruction. To address this, Architect's model reconstruction module was applied to high-confidence predictions from individual tools, rather than from the naïve Bayes-based method (**S16 and S17** Figs). Overall, models constructed from high-confidence DETECT predictions resulted in significantly lower recall compared to either the curated models or UniProt annotations. This is consistent with the idea that high predictive range is an important attribute in an enzyme annotation tool used for model reconstruction;

accordingly, when comparing against model annotations, using high-confidence predictions from either EnzDP or PRIAM instead of DETECT resulted in models with higher recall. At the same time, the resulting recall does not significantly differ from the recall obtained when using predictions from the ensemble method. Furthermore, similar precision is obtained when using EnzDP, PRIAM or the ensemble method when reconstructing *C. elegans* or *N. meningitidis* models; in the case of *E. coli*, significantly better precision is obtained by using predictions from EnzDP instead of the ensemble method (when using the KEGG database). Therefore, based on comparisons of EC annotations derived from curated models, there appears to be no benefit to substituting predictions from tools with high predictive range with those from the ensemble approach. However, when comparing against annotations obtained from UniProt, the ensemble method results in higher recall than EnzDP for all 3 organisms and higher recall than PRIAM for *C. elegans*; similar precision is observed in all organisms, with the exception of lower precision than EnzDP and PRIAM for *C. elegans*. These conflicting results reflect inherent differences in the gold-standard datasets, which in turn may indicate that existing curated models may have potential to be further expanded by using UniProt annotations. Interestingly, use of PRIAM's high confidence EC predictions as input to Architect's reconstruction module results in more accurate annotations than models generated from PRIAM's reconstruction tool (**Fig 3**), highlighting methodological differences between the two tools. Furthermore, unlike the PRIAM pipeline, models constructed by Architect are simulation-ready.

## 2.4 Reaction database and predictive range impact models generated by Architect

Beyond enzyme annotations, we were interested in comparing the performance of models generated by Architect, CarveMe and ModelSEED in terms of simulation results and quality (**Fig 4**). PRIAM-based reconstructions were excluded from these comparisons as they require additional refinements to be used as models of metabolic flux. Further, only models based on the two bacterial species (*N. meningitidis* and *E. coli*) were examined to avoid the potentially confounding influence of assigning reactions to specific subcellular compartments. First, we calculated the performance of the models in terms of gene essentiality predictions. In the subsequent comparisons, precision and recall were computed with reference to gene deletion studies performed *in vivo* in minimal media and under aerobic conditions [19,20]. In general, Architect reconstructions made using BiGG reaction definitions perform at least similarly to models generated by other tools, notably with higher recall compared to CarveMe in *N. meningitidis* and *E. coli*, and higher precision than ModelSEED in *E. coli*. Interestingly, using the KEGG reaction database for Architect reconstructions does not necessarily produce the same results, importantly with lower recall than ModelSEED in *E. coli*. This suggests that the reaction database has an impact on models produced; in particular, the BiGG database consists of reactions present in curated models [35], making it a higher-quality database than KEGG. At the same time, Architect-BiGG's better performance compared to CarveMe, despite their use of the same reaction database, gives merit to Architect's model reconstruction strategy. We also note that, intriguingly, building the *E. coli* CarveMe model without specifying the need to sustain growth in anaerobic conditions gives a model with similar performance to Architect-BiGG (89.6% precision and 48.1% recall), suggesting that the CarveMe methodology may be highly sensitive to the conditions of growth specified for the reconstruction, unlike Architect. In addition to reaction database, the predictive range of the tools used for EC annotation in Architect has some impact, as observed by the lower recall obtained when using high-confidence DETECT predictions only (**S16** and **S17 Figs**).

**Fig 4. Comparison of simulation results of Architect, CarveMe and ModelSEED in (a) *N. meningitidis* and (b) *E. coli*.** Panel (i) shows the performance of each tool in predicting essential genes in minimal media. Yield of ATP and biomass are shown in panels (ii) and (iii) given 1 mmol of glucose and alternate carbon sources in *N. meningitidis*, and under presence and absence of oxygen in *E. coli*. Panel (iv) shows the performance of individual tools in predicting the capacity to grow on alternate media. Error bars show the 95% confidence interval for individual performance measures, each considered as the estimate of a binomial parameter. In those cases, p-values are computed using Fisher's exact test and only between Architect-BiGG or Architect-KEGG and other tools (with *, ** and *** representing *p* less than 0.05, 0.005 and 0.0005 respectively).

Overall, it remains that all the tools generate models with low recall with respect to predicting gene essentiality. This relatively high rate of false negatives may be explained by several factors including: 1) certain essential genes may have been excluded from reconstructed models or misassigned function; 2) key Boolean relationships between multiple genes associated with a single reaction—such as with heteromeric enzyme complexes [5]—may not be captured in the models; or 3) the biomass equation used during model simulations may be incomplete.

Interestingly, of the genes experimentally found to be essential, 71.7% and 76.7% were incorporated into Architect models for *N. meningitidis* and *E. coli* respectively built using BiGG; however, more than half (62.8% and 68.1% respectively) were not predicted to be essential (**S5 Table**), suggesting avenues for improving Architect by either limiting pathways predicted from ECs (thereby reducing pathway redundancy and highlighting the essentiality of certain genes), or through better representations of gene-protein-reaction relationships. Additionally, given that 65 and 94 essential genes are not predicted as essential by Architect or any of the tools in *N. meningitidis* and *E. coli*, despite 80% of the genes being found in the *N. meningitidis* CarveMe model or at least 47% being found in any of the *E. coli* models (**S18 Fig**), all tools may benefit from better predictions of gene-protein-reaction relationships.

## 2.5 Architect models predict growth of *E. coli* in alternate conditions with similar accuracy as other tools

In addition to essential genes, models can be assessed on their predictions of an organism's capacity to grow on alternate media [7,20]. Here, we focussed on evaluating such predictions for *E. coli* for which experimental data generated using the Biolog platform is available [33]. We found that, overall, on the conditions tested, all tools produce models with similar accuracy (**Fig 4**). At the same time, CarveMe achieves perfect recall at the expense of specificity, a consequence of the model's ability to predict growth in all conditions. We note that, unlike in the case of gene essentiality predictions, the specificity remains low (4.3%) with an *E. coli* CarveMe model fitted to grow in aerobic conditions. At the same time, only 23 of the 90 conditions tested via simulations were determined to be non-growth sustaining *in vivo* (**S6 Table**); moreover, we limited our predictions to conditions which could be simulated in all models. Differences in conditions being simulated may explain the inconsistency of our results with those of Machado et al, notably the higher specificity and lower recall of *E. coli*'s CarveMe model [7].

In the absence of large-scale phenotype data for *N. meningitidis*, the sufficiency of 6 carbon sources assessed *in vivo* or *in silico* in Nmb_iTM560 to sustain growth was predicted [19,31]. Notably, given Nmb_iTM560's prediction of no growth only on acetate, the CarveMe model is most in agreement with the curated model, followed by Architect-KEGG, then ModelSEED. Surprisingly, the Architect-BiGG model fails to predict growth on any other carbon source than glucose. This may be partly due to Architect's stringent cutoff for inclusion of non-EC related reactions from the BiGG database. Indeed, all Architect-BiGG reactions are found in the corresponding CarveMe model, and of the 2,723 reactions unique to the CarveMe model, 1,393 would only be included into the Architect model based on high sequence similarity. Notwithstanding these factors, it is important to recall that the CarveMe model is larger than the Architect-based model and comprises the same set of reactions as *E. coli*, which is inconsistent with the findings of Mendum et al [19]. It is thus possible that, on a larger dataset, CarveMe's *N. meningitidis* model might demonstrate low specificity as it may over-predict growth-enabling conditions of growth.

## 2.6 Elimination of energy-generating cycles in universal reaction databases is key for realistic ATP and biomass yields

Next, model quality was assessed by computing the ATP and biomass yields of the *N. meningitidis* and *E. coli* models in aerobic conditions and in both presence and absence of oxygen respectively (**Fig 4**). Such tests are practically useful as unrealistic energy-generating cycles can artificially inflate predicted growth rates [36]. Given literature values of 26 and 2.8–3.2 mmol of ATP produced per mmol of glucose during aerobic and anaerobic growth [37], models

using BiGG reaction definitions (Architect and CarveMe) have biologically realistic ATP yield, with numerically comparable yield for Architect models and expected lower yield in anaerobic conditions for both models. However, models generated using either ModelSEED or Architect with KEGG reaction definitions produce thermodynamically unrealistic ATP yields. Indeed, our simulations reveal that ATP is hydrolyzed even in the absence of glucose. These may be caused in part by incorrect reaction reversibilities causing energy-generating cycles. For instance, after setting as irreversible the reaction catalyzed by 6-phosphogluconolactonase (EC:3.1.1.31 and R02035: D-Glucono-1,5-lactone 6-phosphate + H2O ➔ 6-Phospho-D-gluconate) in the *N. meningitidis* Architect model, only 11 mmol of ATP per mmol of glucose are available to hydrolyze, moreover with no flux through the ATP hydrolysis reaction in the absence of glucose. Additionally, examples of energy-generating cycles have previously been identified in other ModelSEED models [36]. Interestingly, the absence of oxygen does not impact the biomass yield in models reconstructed using ModelSEED and Architect-KEGG, another likely consequence of biologically irrelevant cycles within the models. Overall, this analysis supports the idea that such details as reaction reversibilities within the database used for reconstruction need to be refined. Thus, CarveMe and Architect-BiGG directly benefit from the high-level of curation their reaction database has undergone.

## 3. Discussion

Here, we present Architect, an approach for automatic metabolic model reconstruction. The tool consists of two modules: first, enzyme predictions from multiple tools are combined through a user-specified ensemble approach, yielding likelihood scores which are then leveraged to produce a simulation-ready metabolic model. Through the use of various gold-standard datasets, we have shown that Architect's first module produces more accurate enzyme annotations, and that its second module can be used to produce organism-specific metabolic models with better annotations than similar state-of-the-art reconstruction tools, including CarveMe and PRIAM. Our expectation is that these models serve as near-final drafts, requiring users to perform only minimal curation to incorporate organism-specific data. For example, models for eukaryotic organisms require the independent definition of cellular compartments, currently not supported by Architect.

Improving annotations by combining EC predictions from multiple tools has been previously [38] investigated with performance tested against *E. coli* EC annotations from EcoCyc [39] as gold-standard. Here, the performance when taking a simple union or intersection of annotations from KEGG, RAST, EFICAz and BRENDA (BRaunschweig ENzyme DAtabase) predictions [24,40–42] was evaluated. Either strategy was found to yield either increased precision or recall, but not at the same time. By contrast, when evaluating on *E. coli* EC annotations in SwissProt (**S7 Fig**) and on a larger multi-species set of sequences also from the reference database (**Fig 2**), most ensemble approaches implemented within Architect achieve both increased precision and recall. At the same time, we note the EnzymeDetector, a well-maintained platform within BRENDA which, given the provenance of a sequence's annotation, computes an overall confidence score for enzyme annotations from database-specific weights [43,44]. Significantly, the EnzymeDetector is updated every six months, an important aspect considering that enzyme definitions and sequence information are ever evolving [44]. Therefore, we anticipate that Architect may need to be regularly updated for its reconstructed models to remain relevant as new biochemical information is characterized.

Importantly, our quantitative comparisons of reconstructed metabolic models depend on the quality of the manually reconstructed models being used as benchmark. Regarding organisms challenging to culture, given that their metabolism is therefore difficult to investigate *in*

*vivo*, it is likely that enzymatic functions have been largely computationally inferred during manual reconstruction. On the other hand, while experimental characterization of sequence function is actively being carried out for model organisms like *E. coli*, such organisms are already well-represented in the ENZYME [10] or SwissProt databases [17], possibly biasing Architect's training. To strike a balance, we selected both model organisms as well as lesser studied organisms for our comparison of performance of Architect as a model reconstruction tool.

Interestingly, it is unclear whether improvements in enzyme annotation, other than in terms of predictive range, lead to the construction of models with either improved annotations or greater accuracy of simulations. Instead, we propose three improvements to the input and the algorithm of the model reconstruction module that will likely yield better models. First, we find that most essential genes also incorporated into the final models were not predicted to be essential *in silico* (see **S1 Text**), suggesting that more accurate predictions of gene essentiality may be obtained by better encoding gene-protein-reaction relationships or by limiting the reactions included in the high-confidence model based on EC annotation. For example, while multifunctional enzymes are biologically relevant, it has been suggested that certain ECs associated with multiple reactions may need to be redefined to capture the diversity and specificity of biochemical reactions [45]. Given the prevalence of multi-reaction ECs in databases used by Architect (**S19 Fig**), its indiscriminate inclusion of all reactions associated with high-confidence EC predictions may need to be revisited in a future iteration. Second, enabling the predictions of transport reactions is needed to define the accurate import or export of metabolites which otherwise represent dead-ends in the initial network; in turn, this may lead to fewer blocked or inactive reactions. The comparatively high proportion of blocked reactions in Architect models (**S4 Table** and **S1 Data**) is also partially a consequence of the tool's parsimonious approach to gap-filling given that only reactions required for fulfilling an objective function are considered as gap-fillers; at the same time, the additional reaction and metabolite annotations in Architect's SBML output may guide users in finding other links between blocked reactions and biomass production or other biological processes. Third, considerations of thermodynamics have been absent from our reconstruction pipeline, whether in terms of reaction reversibility, or in terms of gap-filling. Identifying thermodynamically likely solutions for gap-filling is expected to result in more biologically realistic models [46].

Overall, while making any of these improvements, regular MEMOTE checks will also likely be invaluable. For example, running MEMOTE on Architect's SBML output enabled us to identify missing descriptors for reactions, metabolites and genes. This issue could then be relatively simply fixed so as to improve the descriptive quality of models (**Methods** and **S4 Table**). At the same time, the magnitude of MEMOTE scores only partially describes a model's quality. Indeed, the failure of Architect models made using the KEGG database to capture realistic ATP yields conflicts with their higher MEMOTE scores compared to those models made using CarveMe. Therefore, in future, we expect application of such sanity checks as verifying the yield of key metabolites—along with MEMOTE—to further help refine Architect's automatic model reconstruction process.

## 4. Methods

### 4.1 Sources of sequence data

Sequences were downloaded from the SwissProt database [17] and their corresponding annotations from the ENZYME database [10] (downloaded on February 9th, 2021). The ENZYME database is a dedicated database corresponding to enzymatic activity, and is regularly updated mainly following recommendations from the Nomenclature Committee of the International

Union of Biochemistry and Molecular Biology (IUBMB) [47]. Only complete EC numbers were considered in building Architect's ensemble classifiers. Further, ECs associated with fewer than 10 protein sequences were removed to ensure sufficient training data. This filtering resulted in a final collection of 1,670 ECs represented by 207,121 sequences (**S20 Fig**). A further set of 294,067 protein sequences not associated with either complete or partial EC annotations (subsequently referred to as "non-enzymes") was retrieved from the same version of SwissProt. For generation of test and training datasets for use in five-fold cross-validation steps, ECs associated with multifunctional proteins, were divided into appropriate sets using a previously published protocol [48].

## 4.2 Enzyme annotation using ensemble classifiers

For any given protein, EC predictions are generated through integrating the output from five state-of-the-art enzyme annotation tools: EFICAz v2.5.1 [24], PRIAM version 2018 [8], DETECT v2 [14], EnzDP [15] and CatFam [23]. In addition to examining the performance of two relatively simple approaches, majority rule (in which we take the prediction supported by the most tools) and EC-specific best tool (in which we take the prediction from the tool which is found to perform best for a specific EC), we also investigated the performance of the following three classifiers: (1) naïve Bayes, (2) logistic regression and (3) random forest. For training each method, we first find, for each EC $x$, positive (proteins actually of class $x$) and negative examples (other proteins predicted by any tool to have activity $x$). For each protein $i$, a feature vector is then constructed consisting of the level of confidence in each tool's prediction (based on the confidence score output by the tool); the associated binary label $y_i$ indicates whether the $i^{\text{th}}$ protein has activity $x$ and thus has value 1 if and only if the protein has activity $x$. For any EC predicted by all tools without false positives (that is without proteins of other classes misclassified with the EC), we apply a rule whereby we automatically assign the EC if made by any of these tools. Otherwise, those predictions made by an ensemble method with a likelihood score greater than 0.5 are considered to be of high-confidence.

For the naïve Bayes classifier trained on high-confidence predictions, given an EC predictable by $k$ tools ($1 \leq k \leq 5$ depending on the number of tools that can predict the EC), each protein sequence is assigned a corresponding feature vector $F$ of length $k$, where $F_i = 1$ if the EC was predicted with high-confidence by the $i^{\text{th}}$ tool, and $F_i = 0$ otherwise. The posterior probability of a new protein $j$ having EC $x$ (the aforementioned "likelihood score") is then given by the following equation, where each feature is assumed to follow a Bernouilli distribution:

$$p\left(y_j = 1 | F_1 = f_1, \ldots, F_k = f_k\right) = \frac{p(y_j = 1) \cdot \Pi_{i=1}^{k} p(F_i = f_i | y_j = 1)}{\sum_{C \in \{0,1\}} p(y_j = C) \cdot \Pi_{i=1}^{k} p(F_i = f_i | y_j = C)} \tag{1}$$

Other ensemble methods explicitly consider the level of confidence by each tool (see **S1 Text**). For example, our logistic regression classifiers train on feature vectors which use one-hot encoding to denote the level of confidence in the EC prediction by each tool. In the case of our random forest classifiers, each element of the feature vectors takes on a discrete value indicating the level of confidence by each tool.

Given that those ECs associated with fewer than 10 protein sequences are filtered out of the training data, some ECs may not be predictable by the classifier but may nevertheless be predicted by other tools; in particular, PRIAM consists of profiles specific to ECs associated with as few as a single sequence. To ensure higher coverage of metabolic reactions and pathways, EC predictions made with high-confidence by PRIAM are subsequently assigned as high-confidence during downstream model reconstruction.

## 4.3 Reconstruction of metabolic networks

From the set of high confidence enzyme annotations generated for the organism of interest (using the naïve Bayes classifier by default), an initial metabolic model is constructed with reference to either the Kyoto Encyclopedia of Genes and Genomes (KEGG) resource [26] or the Biochemical, Genetic and Genomic (BiGG) knowledgebase [27]. In brief, reaction identifiers and equations corresponding to high-confidence EC predictions are collated, along with non-enzymatic/spontaneous (as indicated in KEGG) and any user-specified reactions (see **S1 Text**). Amongst the latter reactions, an objective function (such as biomass production) is required for downstream gap-filling. Furthermore, if BiGG is used as the reference reaction database, we also include non-EC associated reactions (including transport reactions). This step is performed through BLAST-based sequence similarity searches of the organism's proteome against a database of protein sequences representing these non-EC associated reactions, collated from the BiGG resource, using an E-value cut-off of $10^{-20}$ [11,12].

Having generated an initial network, Architect next attempts to fill gaps within the network, representing reactions required to complete pathways necessary for the production of essential metabolites (as defined by the objective function). First a global set of candidate gap-filling reactions ($R$) is constructed by combining: 1) reactions that were previously identified in the enzyme annotation step at either low- or no-confidence; and 2) exchange reactions for dead-end metabolites (whose presence otherwise results in inactive (blocked) reactions that can inhibit biomass production [49]). From this global set, Architect attempts to identify a set of reactions which, when supplemented to the initial network, is minimally sufficient for non-zero flux through the aforementioned user-defined objective function. This process leverages the mixed-integer linear programming (MILP) formulation employed by the CarveMe pipeline [7]. First, penalties are assigned for the addition of each gap-filling candidate as follows. For the $i^{\text{th}}$ reaction associated with 1 or more ECs predicted with low-confidence by the ensemble classifier ($0.0001 < \text{score} \leq 0.5$), we find the highest score $s_i$ associated with any of the corresponding EC annotations. Then, we scale the scores of the gap-filling candidates to have a median of 1 (where $s_M$ is the median of the original scores):

$$s'_i = \frac{s_i}{s_M} \tag{2}$$

The penalty $p_i$ for adding the $i^{\text{th}}$ reaction is then inversely proportional to the normalized score:

$$p_i = \frac{1}{1 + s'_i} \tag{3}$$

Remaining candidate reactions for gap-filling (that is, those either not predicted with any likelihood score or which are exchange reactions for dead-end metabolites) are each assigned by default a penalty of 1. (Users are also able to specify a higher penalty for exchange reactions for dead-end metabolites, for example, when wishing their model to use only user-specified media.) The following MILP formulation then identifies a subset of reactions from the global set of candidate gap-filling reactions that together have the smallest sum of penalties and enable a minimum production of biomass ($\beta = 0.1$ h$^{-1}$ by default).

$$\text{Minimize} \sum_{i \in R} p_i y_i \tag{4}$$

$$\text{subject to}: \ Sv = 0$$

$$v_L \leq v \leq v_U$$

$$y_i v_{L,i} \leq v_i \leq y_i v_{U,i}, \forall i \in R$$

$$y_i \in \{0, 1\}, \forall i \in R$$

$$c^T v \geq \beta$$

Here, the variables are the flux vector $v$ and $y$. Each $y_i$ serves as an indicator variable: $y_i = 1$ if and only if the $i$th candidate gap-filler (with flux $v_i$) is included in the solution.

## 4.4 Comparisons of annotations and networks

Architect was used to annotate enzymes to the proteomes of three species: *Caenorhabditis elegans*, *Escherichia coli* (strain K12) and *Neisseria meningitidis* (strain MC58) using sequences collated from WormBase (WS235, [50]), SwissProt and the Ensembl database [51] respectively. For each species, the naïve Bayes-based ensemble method was retrained by excluding sequences from the respective organism. Architect predictions were evaluated against gold standard datasets derived from UniProt for *C. elegans* and *N. meningitidis*, and SwissProt for *E. coli*. Performance was reported in terms of specificity and micro-averaged precision and recall (that is, irrespective of enzyme class size) [52].

For network comparisons, models were first generated by Architect, CarveMe v1.2.2, PRIAM v2018 and ModelSEED [7,8]. For *C. elegans*, growth under no specific media was specified and CarveMe's generic biomass function was used for the tool as well as by Architect (with a penalty of 1 for the addition of exchange reactions for deadend metabolites); for ModelSEED, the core template was specified. In the case of *N. meningitidis* and *E. coli*, CarveMe, Architect-BiGG and ModelSEED reconstructions were made using the gram-negative universe defined by each tool. Minimal media was specified, with aerobic and anaerobic conditions specified for *N. meningitidis* and *E. coli* respectively, given the former's inability to grow in strictly oxygen-free conditions [31,32]. Architect reconstructions were made with a penalty of 10 for exchange for deadend metabolites so as to force the models to use the minimal media specified.

These models were evaluated against two sets of gold-standard as described next. Performance was computed using micro-averaged precision and recall first using as a gold-standard, enzyme annotations assigned to genes in previously generated curated metabolic models: *C. elegans*—WormJam [18], *E. coli*—iML1515 [20] and *N. meningitidis*—Nmb_iTM560 [19]. As a second measure of performance, we compared the annotations included in the models following gap-filling against those in UniProt, here restricting the comparison to those ECs present in the relevant reaction database.

## 4.5 Simulation experiments

For *N. meningitidis* and *E. coli*, *in silico* knockout experiments were performed using the models generated by Architect, CarveMe and ModelSEED. Genes predicted to be essential in these *in silico* experiments were subsequently compared to the results of two genome-scale knockout studies [19,20]. Since Architect, unlike CarveMe or ModelSEED, does not predict complex gene-protein-reaction relationships [5], only those reactions associated with a single protein

could be assessed through gene knockout experiments, where the flux through such reactions corresponding to a single protein was constrained to be zero.

Additionally, phenotype array simulations were performed on the *E. coli* models reconstructed using Architect (involving KEGG and BiGG reaction databases), CarveMe and ModelSEED, as was done in [7]. Briefly, each model was first constrained for aerobic minimal media conditions, with glucose, ammonium, sulphate and phosphate as default carbon, nitrogen, sulphur and phosphorus sources respectively. Alternate sources from available Biolog data were selected from [33] based on whether they could be mapped to BiGG, KEGG and ModelSEED compound identifiers and whether they were found in all models being tested. Thus, 90 alternate sources were identified (**S6 Table**). Then, for each type of array, uptake of the default source was blocked and uptake of the alternate source enabled with a maximum bound of 10 mmol/$g_{DW}$/h. We count a prediction as positive for growth when the maximum flux through the biomass function is found to be at least $10^{-4}$ h$^{-1}$. As was done in [33], conditions with readings from the Biolog data marked as 'weak' or 'positive' were considered as positive for growth.

Next, we calculated the biomass yield by first constraining the uptake of glucose or alternate carbon sources (where relevant) to 1 mmol/$g_{DW}$/h. In anaerobic conditions, the uptake of oxygen in each model was constrained to zero. The biomass yield was then obtained by optimizing the flux through the biomass objective function using flux balance analysis. Similarly, the ATP yield was calculated under glucose uptake conditions; the objective function was here set to be the ATP hydrolysis reaction (ATP + $H_2O$ ➔ ADP + phosphate + proton) as was done in [37].

### 4.6 Model output and MEMOTE tests

Simulation-ready Architect models are output in both Excel and SBML Level 3 [16] formats. Following best practices in the systems biology community, expert-curated SBML models as well as those output by Architect, CarveMe and PRIAM were run through the suite of standardized tests offered by MEMOTE v0.13.0 [34]. From these reports (**S1** and **S2 Data**), we improved Architect's codebase by including Systems Biology Ontology terms to each gene, reaction and species element (SBO: 0000243, SBO: 0000375 and SBO: 0000247 respectively). Each Architect SBML model is available at https://github.com/ParkinsonLab/Architect and other models are included in **S1 Data**.

### 4.7 Technical details regarding metabolic simulations

Architect's MILP-based gap-filling is performed using the CPLEX solver (v12.9) and Python v3.7.1. In this paper, other simulations were performed using the Gurobi solver (v6 and v9) and the COBRA Toolbox (v1.3.4 and v3.3) in MATLAB [53].

## Supporting information

**S1 Fig. Performance of different ensemble methods on test set at different levels of sequence identity to the training data.** Comparison of macro-averaged (A) precision, (B) recall, and (C) F1-score of ensemble methods at different MTTSIs—defined in S1 Text. (EPS)

**S2 Fig. Precision and recall of ensemble methods and individual tools for enzyme annotation with respect to ECs predictable by all tools and on different sets of proteins.** (i) Comparison of precision and recall of ensemble (with and without multi-EC filtering) and individual tools on ECs predictable by all tools in (A) the entire test data set, (B) only proteins with a single EC, and (C) only multifunctional proteins. The subscripts "all" and "high" reflect

whether all or only high-confidence predictions from individual tools were considered, respectively. (ii) Comparison of precision and recall of ensemble methods (with and without multi-EC filtering) on ECs predictable by all tools in (A) the entire test data set, (B) only proteins with a single EC, and (C) only multifunctional proteins.
(EPS)

**S3 Fig. Precision and recall of ensemble methods and individual tools for enzyme annotation with respect to ECs predictable by any tool and on different sets of proteins.** (i) Comparison of precision and recall of ensemble (with and without multi-EC filtering) and individual tools on ECs predictable by any tool in (A) the entire test data set, (B) only proteins with a single EC, and (C) only multifunctional proteins. The subscripts "all" and "high" reflect whether all or only high-confidence predictions from individual tools were considered, respectively. (ii) Comparison of precision and recall of ensemble methods (with and without multi-EC filtering) on ECs predictable by any tool in (A) the entire test data set, (B) only proteins with a single EC, and (C) only multifunctional proteins.
(EPS)

**S4 Fig. Performance of different ensemble methods on test set when considering ECs predictable by an increasing number of tools.** Comparison of (A) macro-precision, (B) macro-recall, and (C) F1-score of individual tools and ensemble methods on proteins with ECs predictable by at least 1, 2, 3, 4 tools and by all tools.
(EPS)

**S5 Fig. Class-by-class comparison of performance of the naïve Bayes method and two enzyme annotation tools, DETECT and PRIAM.** (A): Class-by-class comparison of (i) precision and (ii) recall between the naïve Bayes-based ensemble method and DETECT (high-confidence) when considering ECs predictable by DETECT. Each dot represents an EC class, and those dots above the line correspond to EC classes for which the ensemble method performs better. Only EC classes with defined precision from both DETECT and the ensemble classifier are shown. (B) Same as (A), except that the class-by-class comparison is between the naïve Bayes-based ensemble method and PRIAM (high-confidence predictions) and on ECs predictable by PRIAM. Again, only EC classes with defined precision from both PRIAM and the ensemble classifier are shown.
(EPS)

**S6 Fig. Specificity of individual tools (high-confidence) and ensemble approaches on non-enzymatic dataset.**
(EPS)

**S7 Fig. Comparison of organism-specific performance of Architect's enzyme annotation tool using predictions by the naïve Bayes ensemble method and three individual tools, DETECT, EnzDP and PRIAM.** High-confidence PRIAM predictions were included to the predictions of the naïve Bayes classifiers for those ECs outside of Architect's predictive range. The comparison was done over those proteins present in both the input protein sequence file and those found in UniProt/SwissProt. Error bars show the 95% confidence interval for precision, recall and specificity, each considered as the estimate of a binomial parameter. P-values are calculated using Fisher's exact test and between Architect results and other tools only (with *, ** and *** representing $p$ less than 0.05, 0.005 and 0.0005 respectively).
(EPS)

**S8 Fig. Comparison of performance of the naïve Bayes method on the test set when training on predictions from combinations of fewer than 5 tools.** In (A), the light green square in

each row indicates when a tool's predictions are being considered in a combination; this table shows the combinations of tools ranked from highest F1-score to lowest amongst those involving 2, 3 and 4 tools respectively. In (B), the purple diamond indicates the performance when the naïve Bayes-based method is trained on the entire set of predictions, whereas each green circle indicates performance when considering a particular combination of tools (number corresponding to the rank in (A)).
(EPS)

**S9 Fig. Overlap of annotations in models reconstructed with Architect (based on KEGG), CarveMe, PRIAM's reconstruction tool and ModelSEED using as gold-standard (i) curated models, and (ii) UniProt/SwissProt.** In comparison against model annotations (i), Architect's higher precision and recall in (A) *C. elegans* are due to having both more annotations in common with WormJam (that is, more true positives and fewer false negatives) and fewer predictions unique to the tool (that is, false positives). In *N. meningitidis* (B)(i), Architect's higher precision against PRIAM and CarveMe are due to its lower ratio of false positives to true positives with Nmb_iTM560. ModelSEED however demonstrates better performance by virtue of predicting more true positives and fewer false positives. CarveMe's better performance in *E. coli* (C)(i) can be attributed to its annotations intersecting more with those in *i*ML1515 than Architect; on the other hand, ModelSEED's performance benefits from its fewer false positive predictions. Either of these may be due to the presence of *E. coli*-relevant data in CarveMe and ModelSEED databases. Interestingly, despite the use of PRIAM annotations by Architect—explaining the high-level of commonality between annotations produced by both—the latter in general has statistically higher performance due to higher ratios of true positives to false positives (in all 3 organisms) and to false negatives in *C. elegans*. In comparison against UniProt/SwissProt (ii), more of Architect's annotations constitute true positives versus in comparison to the organism's models, suggesting that the existing models for *C. elegans*, *N. meningitidis* and *E. coli* may be improved through consideration of annotations from the database. Overall, based on our comparisons, Architect shows itself to be complementary to the other tools, supplementing annotations missed by them.
(EPS)

**S10 Fig. Number of pathway-specific ECs present in Architect's KEGG database versus in CarveMe's main BiGG database.** Each dot represents a pathway in KEGG, and the diagonal line indicates the points on the graph where a pathway is covered by the same number of ECs through KEGG and BiGG.
(EPS)

**S11 Fig. Performance of Architect as a model reconstruction tool (using BiGG as the reaction database) versus CarveMe in the case of the three organisms of interest.** Quality of annotations is computed against the curated models over the genes found in these models, and against UniProt/SwissProt when restricting to those sequences found in the database and with ECs present in the BiGG reaction database. Error bars show the 95% confidence interval for precision and recall, each considered as the estimate of a binomial parameter. P-values are calculated using Fisher's exact test (with *, ** and *** representing p less than 0.05, 0.005 and 0.0005 respectively).
(EPS)

**S12 Fig. Overlap of annotations in models reconstructed with Architect (based on BiGG) and CarveMe using as gold-standard (i) curated models, and (ii) UniProt/SwissProt.** In comparisons for *C. elegans* (A) and *N. meningitidis* (B), fewer predictions non-overlapping with either gold-standard are made by Architect alone as compared to CarveMe. In *E. coli* (C),

while more of CarveMe's annotations intersect with those in *i*ML1515, similar ratios of true positive to false positive yield similar precision. However, in comparison against SwissProt (C) (ii), higher ratios of true positive to false positive or false negative result in Architect having better performance. As suggested in comparisons involving Architect models made with KEGG reaction definitions, Architect annotations may be used to complement models made using CarveMe.
(EPS)

**S13 Fig. Performance of Architect at varying likelihood score cutoffs (using KEGG reaction definitions).** Average precision and recall over 20 models reconstructed with Architect (using KEGG reaction definitions) when using the default cutoff of 0.5 for high-confidence predictions are indicated, along with averaged performance when using cutoffs of 0.1, 0.3, 0.7 and 0.9. Precision and recall are calculated in terms of annotations (given curated metabolic models of individual organisms and SwissProt/Uniprot) and essential genes predicted in minimal media. Error bars indicate standard deviations.
(EPS)

**S14 Fig. Performance of Architect at varying likelihood score cutoffs (using BiGG reaction definitions).** Average precision and recall over 20 models reconstructed with Architect (using BiGG reaction definitions) when using the default cutoff of 0.5 for high-confidence predictions are indicated, along with averaged performance when using cutoffs of 0.1, 0.3, 0.7 and 0.9. Precision and recall are calculated in terms of annotations (given curated metabolic models of individual organisms and SwissProt/Uniprot) and essential genes predicted in minimal media. Error bars indicate standard deviations.
(EPS)

**S15 Fig. Performance of PRIAM for draft model reconstruction at varying probability cutoffs.** Precision and recall when using the default cutoff of 0.5 are indicated, along with results of using cutoffs of 0.1, 0.2, 0.3, 0.7 and 0.9. Precision and recall are calculated in terms of annotations (given curated metabolic models of individual organisms and SwissProt/Uniprot).
(EPS)

**S16 Fig. Performance of Architect as a model reconstruction tool (with the KEGG reaction database) when using as input high-confidence predictions from the naïve Bayes-based method or from DETECT, EnzDP or PRIAM.** Quality of annotations is computed against the curated models over the genes found in these models, and against UniProt/SwissProt when restricting to those sequences found in the database and with ECs present in the KEGG reaction database. Genes determined essential *in silico* are compared against those whose knockout was tested *in vivo*. Error bars show the 95% confidence interval for precision and recall, each considered as the estimate of a binomial parameter. P-values, computed using Fisher's exact test, are calculated only between Architect run on the ensemble method and on Architect run on any of the individual tools' predictions (with *, ** and *** representing *p* less than 0.05, 0.005 and 0.0005 respectively).
(EPS)

**S17 Fig. Performance of Architect as a model reconstruction tool (with the BiGG reaction database) when using as input high-confidence predictions from the naïve Bayes-based method or from DETECT, EnzDP or PRIAM.** Quality of annotations is computed against the curated models over the genes found in these models, and against UniProt/SwissProt when restricting to those sequences found in the database and with ECs present in the BiGG reaction database. Genes determined essential *in silico* are compared against those whose knockout was

tested *in vivo*. Error bars show the 95% confidence interval for precision and recall, each considered as the estimate of a binomial parameter. P-values, computed using Fisher's exact test, are calculated only between Architect run on the ensemble method and on Architect run on any of the individual tools' predictions (with *, ** and *** representing *p* less than 0.05, 0.005 and 0.0005 respectively).
(EPS)

**S18 Fig. Overlap with experimentally validated essential genes with those predicted essential by Architect-KEGG, Architect-BiGG, CarveMe and ModelSEED for (a) *N. meningitidis* and (b) *E. coli* in minimal media.** For genes completely missed by all tools, the numbers within brackets indicate how many are present in the Architect-KEGG, Architect-BiGG, CarveMe and ModelSEED models respectively.
(EPS)

**S19 Fig. Frequency of ECs associated with 1 or multiple reactions in the KEGG and main BiGG reaction databases used by Architect.**
(EPS)

**S20 Fig. Properties of ECs and proteins included in Architect's database.** (A) Proportion of ECs in Architect's training and test datasets combined associated with different numbers of proteins, with proportions relevant to the subset of those ECs predictable by all tools indicated in grey. The percentages of ECs predictable by any tool and associated with different numbers of proteins are indicated above the bars. (B) Intersection of ECs predictable by different tools, when focussing on ECs (i) found in Architect's training database and (ii) more generally. These Venn diagrams were made using the interface at *http://bioinformatics.psb.ugent.be/ webtools/Venn*. (C) Number of proteins in the training and test sets combined associated with 1 or more ECs. The percentage of the proteins involved in each category is given above the bars.
(EPS)

**S1 Table. Breakdown of annotations of SwissProt sequences by individual and ensemble methods into true positives, true negatives, false positives and false negatives.**
(DOCX)

**S2 Table. Breakdown of organism-specific annotations by ensemble and individual tool into true positive, false positive and false negative.**
(DOCX)

**S3 Table. Comparisons of various aspects of model reconstruction for *C. elegans*, *N. meningitidis* and *E. coli*.**
(DOCX)

**S4 Table. Summary of MEMOTE results for models reconstructed using Architect (KEGG and BiGG universes), CarveMe, PRIAM and ModelSEED and those manually reconstructed.**
(DOCX)

**S5 Table. Overlap of *in silico* determined essential genes with those found essential *in vivo*.**
(DOCX)

**S6 Table. Details of *E. coli* phenotypes tested *in silico*.**
(DOCX)

**S1 Data. SBML models of *C. elegans*, *E. coli* and *N. meningitidis* built using various reconstruction tools.** Architect models built using BiGG and KEGG reaction definitions are found in the folders named Architect_BiGG and Architect_KEGG respectively, and those built using CarveMe, ModelSEED and PRIAM are similarly found in the folders thus named. Where applicable, additional settings had to be specified as follows. *C. elegans* models were built using undefined/complete media and ModelSEED's core template and BiGG's main reaction database (in the case of CarveMe and Architect-BiGG reconstructions). *E. coli* and *N. meningitidis* models were built under specifications of anaerobic and aerobic (respectively) minimal media conditions and with the specification that a gram-negative organism's sequences had been input.
(ZIP)

**S2 Data. MEMOTE reports for SBML files corresponding to manually curated and automatically reconstructed models of *C. elegans*, *E. coli* and *N. meningitidis*.** The reports of the manually curated models are found in the folder named Benchmarks, and those reports detailing statistics of the automatically generated models are located in the folders named according to the tool utilized. MEMOTE v0.13.0 was used to generate the reports.
(ZIP)

**S1 Text. Additional details about Architect.** Individual sections concern (A) individual enzyme annotation tools, (B) ensemble approaches used to combine predictions from multiple tools, (C) details of performance metrics, (D) metabolic model reconstruction and simulation details, (E) specific parameters specified for metabolic reconstruction by different tools, and (F) additional results about a heuristic for filtering multifunctional enzyme predictions.
(DOCX)

## Acknowledgments

We thank members of the Parkinson and Moses labs for constructive feedback throughout the development of Architect. We also thank Dr Swapna Seshadri, Billy Taj and Dr Xuejian Xiong for their advice in code development. In addition, we thank Andrew Duncan, Shraddha Khirwadkar and Dr Xuejian Xiong for testing Architect, and Ana Popovic for valuable feedback on building Architect's manual.

## Author Contributions

**Conceptualization:** Alan M. Moses, John Parkinson.

**Data curation:** Nirvana Nursimulu.

**Formal analysis:** Nirvana Nursimulu.

**Funding acquisition:** John Parkinson.

**Investigation:** Nirvana Nursimulu.

**Methodology:** Nirvana Nursimulu, John Parkinson.

**Project administration:** Alan M. Moses, John Parkinson.

**Resources:** John Parkinson.

**Software:** Nirvana Nursimulu.

**Supervision:** Alan M. Moses, John Parkinson.

**Visualization:** Nirvana Nursimulu.

                                    

**Writing – original draft:** Nirvana Nursimulu, John Parkinson.

**Writing – review & editing:** Nirvana Nursimulu, Alan M. Moses, John Parkinson.

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
