## [Decision Letter · Decision Letter 0]

29 Mar 2022

Dear Dr. Parkinson,

Thank you very much for submitting your manuscript "Architect: a tool for producing high-quality metabolic models through improved enzyme annotation" for consideration at PLOS Computational Biology.

As with all papers reviewed by the journal, your manuscript was reviewed by members of the editorial board and by several independent reviewers. In light of the reviews (below this email), we would like to invite the resubmission of a significantly-revised version that takes into account the reviewers' comments. Both reviewers recognized that the gene annotation part of the paper would make a nice contribution; however, they expressed significant shortcomings as to whether the methodology can indeed generate draft metabolic reconstructions. Many aspects of metabolic models and quality checks present in existing tools are absent here. we believe that a re-framed paper that focuses on annotation tasks would fair better. The authors are encouraged to revise the manuscript accordingly and address all points raised by the reviewers.

We cannot make any decision about publication until we have seen the revised manuscript and your response to the reviewers' comments. Your revised manuscript is also likely to be sent to reviewers for further evaluation.

Sincerely,

Jason Papin

Editor-in-Chief

PLOS Computational Biology

Reviewer's Responses to Questions

**Comments to the Authors:**

Reviewer #1: Architect: a tool for producing high-quality metabolic models through improved enzyme annotation

Here the authors present “Architect”, a new automated tool for building genome-scale metabolic models. It differs from other tools by using an ensemble-based enzyme annotation strategy, applying multiple annotation algorithms and using a variety of strategies to combine the resulting annotations into a single ideal annotation selected by their algorithm. The approach displays extremely high recall and precision compared with other methods.

Overall, it’s clear the annotation approach proposed works very well, as least for the model genomes studied (and for swissprot proteins). Of course, EC alone is not always a perfectly precise description of function, as often many reactions will map to a single EC, and not all are always correct.

The idea of using multiple annotation algorithms as an ensemble to arrive at a better model is not entirely novel and has been described before, and the literature was not cited in the present manuscript. See “Combining multiple functional annotation tools increases coverage of metabolic annotation” as an example.

Undeniably, this manuscript presents a wonderful algorithm for genome annotation, which appears to be a significant advance to the state of the art in genome annotation. And if presented strictly on that basis, then this manuscript would be quite suitable and complete, although perhaps not an ideal fit for PLOS Computational Biology. However, the authors present this as a pipeline for automated model reconstruction, which sets the bar much higher. And unfortunately, on that basis, the present manuscript lacks many important details and explanations.

1.) How are biomass reactions predicted? The methods don’t appear to provide details on this. Can the authors supply this information? It seems like this is something that should be in the main text (not just supplement).

2.) How do models perform on typical model-based quality measures (e.g. number of gap filled reactions, number of genes in the model, number of blocked reactions)?

3.) KEGG in particular contains many reactions that are not suitable for modeling (e.g. reactions involving generic compounds like accepter and donor, strange polymer reactions with incorrect stoichiometry, mixtures of differing levels of stereochemical detail on compounds). Did the authors filter these reactions out? What about charging and proton stoichiometry, which is critical for model accuracy (and typically lacking from KEGG)? Did the authors include this information? What about compartmentation and organelles (KEGG does not specify compartments even in transport reactions)? Do the KEGG based models have transporters?

4.) How do models perform on energetic accuracy? Do they overproduce ATP? Information about how biomass and ATP yields compared to other models would be helpful. Again, this is an incredibly important yet challenging aspect of model reconstruction.

5.) How do models perform on predicting phenotype data? Authors imply essentiality predictions should be improved due to superior annotations, but was this tested?

6.)What about GPR associations? Do the models include protein complex information in GPRs, and if so, how is this done?

The manuscript appears to contain little detail on these important questions relating to model quality, unless perhaps this information is buried in supplemental material. Of course, much information must always be put in the supplement, but a summary in the main text highlighting that the points above have at least been considered would be most helpful.

It is important to note that a modeling pipeline does not have to be perfect or do all of the things described above to be useful. This pipeline will be valuable on the basis of its improved annotations alone. However, we should be cautious about implying that a pipeline produces higher quality models than competing pipelines without considering the full measure of a model’s quality, which is more than just the annotation accuracy or even MEMOTE score. MEMOTE is wonderful and a great advance as a metric, but it’s not the sole measure of a model’s quality.

Reviewer #2: PCOMPBIOL-D-22-00131

The manuscript Architect: a tool for producing high-quality metabolic models through improved enzyme annotation by Nursimulu et al. compiles different annotation tools to improve the gene annotation to aid the reconstruction of genome scale metabolic models. Despite some minor wording problems, the manuscript is well-written, and authors do a great job explaining the Architect pipeline. Since the manuscript focuses on annotation, I will recommend changing the name to “Architect: a tool for aiding the reconstruction of high-quality metabolic models through improved enzyme annotation”. The reason of the change is that a tool to produce metabolic models will need to consider:

Addition of constraints to simulate grow under different conditions. Evaluation of accuracy, sensitivity, specificity of predicted growth phenotypes.

Explanation of why authors the selected nematode and two-gram negative bacteria as show case. Does authors tool have a bias?

In the case of the nematode reconstruction, authors should clarify if their tool has any potential to indicate reaction/enzyme localization at organelle level. Development of tools for eukaryotic organisms require a comparison with tools such as Plant-SEED.

Architect seems to be a great support tool during the reconstruction process. However, further explanation of supplementary Figure 9 that shows Venn Diagrams comparing Architet-K with CarveMe and PRIAM is needed. Authors should discuss the possible source of predicted differences by tool (e.g. example, blast parameters)

A comparison of Architect with automatic reconstruction tools such as PATRIC is also needed.

Minor

Abstract. Is missing some stats of the comparative analysis with other reconstruction tools

Line 64. PRIAM is not a reconstruction tool

Line 13. Change constraints by constraint.

Line 37. Rephrase the sentence “-can further …”

Line 40. Change “on a” by “at”

**Have the authors made all data and (if applicable) computational code underlying the findings in their manuscript fully available?**

Reviewer #1: Yes

Reviewer #2: Yes

PLOS authors have the option to publish the peer review history of their article (what does this mean?). If published, this will include your full peer review and any attached files.

Reviewer #1: No

Reviewer #2: No
---

## [Decision Letter · Decision Letter 1]

29 Jul 2022

Dear Dr. Parkinson,

We are pleased to inform you that your manuscript 'Architect: a tool for aiding the reconstruction of high-quality metabolic models through improved enzyme annotation' has been provisionally accepted for publication in PLOS Computational Biology.

Best regards,

Jason Papin

Editor-in-Chief

PLOS Computational Biology

Reviewer's Responses to Questions

**Comments to the Authors:**

Reviewer #1: This reviewer thanks the authors for their very thorough response to all of my initial comments. The extensive changes to the manuscript, combined with the title change, satisfy my concerns about this manuscript for publication in PLOS Computational Biology.

Reviewer #2: Authors have addressed all my comments and suggestions successfully. However, I stumbled into a manually curated reconstruction of Caenorhabditis elegans. As a last comments before publication, authors should include a comparison of Architect with the model published by Wang et. al., https://www.pnas.org/doi/10.1073/pnas.2102344118

**Have the authors made all data and (if applicable) computational code underlying the findings in their manuscript fully available?**

Reviewer #1: Yes

Reviewer #2: Yes

PLOS authors have the option to publish the peer review history of their article (what does this mean?). If published, this will include your full peer review and any attached files.

Reviewer #1: No

Reviewer #2: No

---

## [Editor Report · Acceptance letter]

27 Aug 2022

PCOMPBIOL-D-22-00131R1 

Architect: a tool for aiding the reconstruction of high-quality metabolic models through improved enzyme annotation

Dear Dr Parkinson,

I am pleased to inform you that your manuscript has been formally accepted for publication in PLOS Computational Biology. Your manuscript is now with our production department and you will be notified of the publication date in due course.

With kind regards,

Agnes Pap
